# Challenges and Innovations in Osteochondral Regeneration: Insights from Biology and Inputs from Bioengineering toward the Optimization of Tissue Engineering Strategies

**DOI:** 10.3390/jfb12010017

**Published:** 2021-02-27

**Authors:** Pedro Morouço, Cristiana Fernandes, Wanda Lattanzi

**Affiliations:** 1ESECS, Polytechnic of Leiria, 2411 Leiria, Portugal; 2CDRSP, Polytechnic of Leiria, 2430 Marinha Grande, Portugal; cristiana.fernandes@ipleiria.pt; 3Department of Life Science and Public Health, Università Cattolica del Sacro Cuore, 00168 Rome, Italy; wanda.lattanzi@unicatt.it

**Keywords:** hyaline cartilage, subchondral bone, tissue engineering, regenerative medicine, biomaterials, additive manufacturing

## Abstract

Due to the extremely high incidence of lesions and diseases in aging population, it is critical to put all efforts into developing a successful implant for osteochondral tissue regeneration. Many of the patients undergoing surgery present osteochondral fissure extending until the subchondral bone (corresponding to a IV grade according to the conventional radiographic classification by Berndt and Harty). Therefore, strategies for functional tissue regeneration should also aim at healing the subchondral bone and joint interface, besides hyaline cartilage. With the ambition of contributing to solving this problem, several research groups have been working intensively on the development of tailored implants that could promote that complex osteochondral regeneration. These implants may be manufactured through a wide variety of processes and use a wide variety of (bio)materials. This review aimed to examine the state of the art regarding the challenges, advantages, and drawbacks of the current strategies for osteochondral regeneration. One of the most promising approaches relies on the principles of additive manufacturing, where technologies are used that allow for the production of complex 3D structures with a high level of control, intended and predefined geometry, size, and interconnected pores, in a reproducible way. However, not all materials are suitable for these processes, and their features should be examined, targeting a successful regeneration.

## 1. Introduction

In the last years, the world has been witnessing the progressive increase in the prevalence of debilitating disorders affecting osteochondral tissues, leading to the functional impairment of synovial joints and severe pain [1]. In particular, osteoarthritis (OA) represents a significant health burden in developed and developing countries, mostly due to aging and to the increase of risk factors, including obesity and sedentary lifestyle, along with intervening joint injuries [2,3]. OA is a chronic and etiologically heterogeneous joint disorder, which represents the most prevalent musculoskeletal disorder worldwide. In OA, the progressive degeneration of the hyaline cartilage that lines the surfaces of bones articulating through synovial (i.e., moving) joints causes direct bone-to-bone attrition during movements throughout the body [4,5].

The joint degenerative process in OA is characterized by an activation of maladaptive repair responses including pro-inflammatory pathways of innate immunity, commonly initiated by micro-and macro-injuries that induce cell stress and extracellular matrix (ECM) degradation [6]. This disease is on the World Health Organization’s top list of concerns, as the breakdown of articular cartilage is a major health matter to which there are few effective solutions, at least with guaranteed success [7]. Direct trauma, chronic degeneration (i.e., mechanical overload), or an abnormality of the underlying subchondral bone are the main detailed factors to produce articular cartilage lesions [8,9]. OA is typically associated with risk factors that reflect lifestyle attitudes, such as aging, obesity, nutritional deficiencies, and physical (in)activity, along with joint-related co-morbidities, including intervening traumas and/or ischemic injuries, malalignment, and abnormal mechanical load [2], interacting in a complex pathophysiology, and causing over 250 million individuals to be affected worldwide. This prevalence justifies the need to improve and boost research efforts that contribute to the development of novel treatments, leveraging integrated and multidisciplinary approaches.

Overall, the osteochondral tissue is an integrated load-bearing structure that represents a significant challenge in regenerative medicine, owing to its nanostructural complexity, stratified architecture, and crucial biomechanical properties [10,11,12]. Hence, engineering the optimum osteochondral construct has been somewhat hampered due to poor tissue formation and problematic integration at the cartilage-bone interface [13].

This review aimed to provide an up-to-date overview of the challenges posed by osteochondral regeneration and an updated categorization of the tissue engineering strategies described in the extant scientific literature.

## 2. Overview of the Joint Surface Structure

The anatomical and histological architecture of bone and (hyaline) cartilage, and the resulting bone-cartilage boundary in long bones, underlies the complexity of this tissue microenvironment and its developmental paths.

### 2.1. Cartilage Structure and Functional Properties

Cartilage is found in the human body in three different tissue subtypes: hyaline cartilage, fibrous cartilage, and elastic cartilage. The hyaline cartilage (also known as articular cartilage) is a flexible connective tissue that aligns the surface of the bones in the synovial joints throughout the body, allowing a movement with almost zero friction on its surface. It has extraordinary mechanical properties (elastic modulus of ~123 MPa; mechanical tensile strength of 17 MPa; compressive modulus between 0.53–1.82 MPa; compressive stress between 14–59 MPa) [14,15,16] and lasting durability, despite having a thickness of only a few millimeters.

The cartilage tissue is poorly cellularized, comprising only one cell type (chondrocytes), which makes up around 2% of the entire volume. In hyaline cartilage, the chondrocytes are clustered in “cell-nests” or “isogenous groups”, submerged in an abundant ECM enriched in water (see Figure 1) [17,18,19,20]. The hyaline cartilage ECM is stratified into four distinct (architecturally and biochemically) zones, namely, the surface zone, the mid zone, the deep zone, and the calcified zone, all contributing to the viscoelastic properties (see Figure 1) [21]. This unique structure and composition provide joints with a surface that combines low friction with high lubrication, shock absorption, and wear resistance while bearing large repetitive loads throughout a person’s lifetime [4,22,23]. The ECM is mostly composed of collagens (mostly type II) and negatively charged proteoglycans, plus “adhesive” glycoproteins and elastin fibers in minor proportions [5,22]. Collagens make up 60% of the dry weight of the cartilage, forming fibrils crosslinked in fibers, interwoven with proteoglycans. The proteoglycans featured in the articular cartilage include aggrecans as the most abundant, and, due to their high affinity with water, they confer resistance to compressional forces by a swelling pressure across the articular joint. This biochemical composition combines tensile strength, provided by collagen fibers, with deformability, provided by aggrecan that binds with water. Hence, cartilage may act as a sponge, releasing water when bearing a mechanical load and resorbing it when the load is reduced. Despite the low metabolic activity, the relatively poor self-healing properties of chondrocytes, and the absent vascularization of the tissue, articular cartilage is considered a dynamic and responsive tissue, in which the contribution of cell-produced ECM components has a noteworthy role [9,23,24,25,26].

The cartilage tissue is devoid of any vascular support (either blood or lymphatic) and of innervation, which explains the very low endogenous regenerative capacity [27]. Chondrocytes are indeed extremely specialized cells that have the unique property of surviving in the low oxygen tension environment of avascular tissue—extreme conditions that are not tolerated by any other cell type in the body. The needed oxygen and nutrients are supplied by a combination of diffusion and fluid flow through the ECM, which is dynamically sustained by joint loading during movements [15]. Chondrogenic precursors are believed to be present exclusively in the perichondrium, i.e., the layer of dense irregular connective tissue that surrounds the cartilage of developing bones. Hence, hyaline cartilage is defined as a terminally differentiated tissue, which lacks any intrinsic plasticity and regenerative capacity, as it cannot replace its cells upon tissue loss or damage. Nonetheless, cartilage may seem a suitable target for regenerative strategies, owing to the capacity to survive without the need to support neo-angiogenesis and to the relative simplicity of its cytological architecture (i.e., a single differentiated cell type).

Taken together, the lack of blood combined with the limited proliferation potential of chondrocytes limit the intrinsic healing process, by inhibiting the transport of inflammatory mediators (both molecules and cells) to the defect site. Although the synthesis rate of glycosaminoglycan (GAG) and collagen (primarily type II) in a developing tissue ex vivo depends on gas exchange, cells remain viable under hypoxic conditions, which are inherent for adult cartilage in an articular joint [28]. Collagen fibrils combined with GAGs provide tensile strength, load-bearing capabilities, and resilience [5]. Either way, their structure and mechanical properties allow them to handle repetitive load forces over decades. Thus, damage caused to the joints by trauma or disease usually requires an exogenous intervention to stimulate regeneration [1,29].

### 2.2. Bone Structure and Functional Properties

The subchondral bone structure is completely different, composed of concentric lamellar layers around osteons and flat layers representing new bone formation [30]. Mature *bone* is vascularized and houses a complex stem cell niche within the bone marrow cavities, comprising the multipotent “mesengenic” stem cell population (namely, mesenchymal stromal cells, MSCs, according to currently approved nomenclature) (see Figure 1). MSCs are located around vessels in the stroma and the endosteal cambium layer and maintain an intense crosstalk with hematopoietic cells [31]. MSCs are naturally able to differentiate toward mesodermal lineages found within a skeletal segment (bone, cartilage, fat, vessels), and are naturally prone to commit toward the osteoblast/chondroblast lineages [32]. Osteogenic precursors are then also found in the inner layer of the periosteum [31]. These properties make the bone naturally capable to regenerate following traumatic injury.

The bone matrix is mineralized due to the unique ECM-production activity of osteoblasts. Mature osteoblasts secrete an organic matrix, comprising dense collagen layers (mostly made up of type I collagen) that alternate in opposite directions that depend upon the stress loading axis. Thereafter, osteoblasts deposit the mineral compound, mainly formed by calcium and phosphate combined in hydroxyapatite (Ca_10_(PO_4_)_6_(OH)_2_) microcrystals (HA). HA then undergoes nucleation and becomes deeply incorporated into the collagen layers [33]. This structure makes the bone matrix stiff and dense to sustain tensile and compression forces; nonetheless, the entire architecture remains extremely plastic. As the bone grows, osteoblasts remain incorporated into the mineralized ECM as *osteocytes*, while new osteoblasts are recruited from MSCs. Despite appearing functionally quiescent, osteocytes are still active in signaling and govern bone homeostasis, by exerting endocrine functions, sensing chemical and mechanical stimuli, and communicating with each other and with the surface through cell processes that run within canaliculi through the mineralized matrix [34,35]. Finally, osteoclasts, large multinucleated cells originating from the bone marrow monocyte lineage, are involved in bone resorption by lowering the pH through proton pump activation and by secreting acidic hydrolases that disrupt the organic matrix [36].

The vascularized nature of bone allows for the direct nourishment of resident cells (i.e., osteocytes, osteoblasts, and osteoclasts), as well as the indirect nourishment of the overlying cartilage layer through paracrine diffusion through the ECM [37]. In particular, the peripheral bone facing the joint cartilage is relatively avascular, as it lacks the periosteum that lines the remainder bony surfaces. The subchondral bone hence serves as an anchorage for the adjacent cartilage collagen fibrils and plays an important role in the maintenance of the joint.

### 2.3. Osteochondral Developmental Insights

Both cartilage and bone originate from the mesoderm germ layer of the embryo (apart from selected skull bone regions that originate from the neural crest), starting at the end of the fourth week of embryo development. At this stage, multipotent stem cells residing in the mesenchyme (mesenchymal stem cells, MSCs) either merge to organize membranes around newly formed visceral cavities or start committing into chondrocytes, forming the early skeletal buds of membranous flat bones and endochondral long and short bones, respectively.

In *membranous bones*, MSC condenses and differentiates into osteoblasts, without any cartilage intermediate; osteoblasts start organizing osteoid lamellae and depositing a mineralized matrix comprising hydroxyapatite, type-I collagen, and other bone-specific ECM compounds (see, for a detailed review, [38]). In *endochondral bones*, MSC condenses and differentiates into chondrocytes that start producing their specific ECM. Subsequently, chondrocytes undergo terminal differentiation to hypertrophic chondrocytes that can mineralize the ECM [39]. The degenerating calcified cartilage is progressively replaced by bone tissue, starting at the fifth week of embryo development, through a process known as “endochondral ossification”, which includes vascular invasion of the cartilage bud and the consequent colonization and repopulation by osteogenic precursors and osteoblasts.

In long bones (taken as best explicative examples) the process of endochondral ossification occurs first in the primary ossification centers in the diaphysis, then in the secondary ossification centers in the epiphyses. The cartilage will persist after birth and until puberty in the epiphyseal plates of long bones to enable bone elongation. After puberty, the process achieves its completion, and the epiphyseal plates are completely ossified, turning into epiphyseal lines. In the adult post-pubertal skeleton, hyaline cartilage remains exclusively in selected regions: articular surfaces in synovial joints, synchondroses, skull base, and sternal ends of ribs.

### 2.4. Key Molecular Signaling in Osteochondral Development and Growth

The correct and chronologically organized fulfillment of these events is determined by a finely tuned molecular control, whose key actors are quite well characterized. The sex-determining region Y (SRY)-box 9 (SOX9) transcription factor is the key player in the initiation stage of cartilage formation, regulating MSC commitment into chondrocytes; Sox9 knockout mice are indeed unable to form cartilage [40]. SOX5 and SOX6 contribute to the further stages, enhancing SOX9 efficacy and specificity, while other molecular mediators contribute to the progression of the entire process [41]. SOX9 is also needed to prevent the conversion of proliferating chondrocytes into hypertrophic chondrocytes, which would then lead to endochondral ossification [40]. The Runt-related transcription factor 2 (RUNX2), also known as core-binding factor subunit alpha-1 (CBFa1), is instead considered the master bone transcription factor, being the very first molecule required for the determination of the osteogenic lineage. RUNX2 suppresses the proliferation of MSCs and induces their differentiation into immature osteoblasts by triggering the expression of major bone matrix genes (osteopontin, osteocalcin, alkaline phosphatase, etc.). Conversely, Runx2, *per se*, inhibits further osteoblast maturation and mature bone formation, and it is not required for maintaining bone genes’ expression in later stages [42]. Furthermore, the TGFβ/BMP and WNT signaling pathways are the main contributors in the osteogenic gene network of signaling cascades (for more details, see [43,44,45,46]).

The alternative molecular paths governing the osteogenic and chondrogenic fates of MSC reflect the inherent differences between these two specialized connective tissues, which can be schematized into three main features: nature of the ECM, vascularization, and cellular composition.

## 3. Tissue Engineering for Osteochondral Regeneration

The regeneration of articular cartilage resulting from degenerative joint disease is an exciting area under investigation using Tissue Engineering (TE) approaches, which have been yielding promising experimental regenerative strategies [47,48]. TE approaches involve the combination of three main components, i.e., scaffold, biomolecules (e.g., growth factors and bone-inducing agents, drugs, such as antimicrobials, etc.), and cells.

Currently, there are several available strategies for treating a joint cartilage lesion, including some already on the market (palliative; microfraction; grafts; cell-based; whole tissue transplantation; scaffold-based) and others that are still under investigation (cell plus scaffold-based; and hydrogels-based or cell plus hydrogel-based). Accordingly, we might question why OA is a burden for health systems. Most of the existing procedures show significant drawbacks and need to be improved to achieve functional osteochondral regeneration. In this regard, most of the available studies test cartilage regeneration under static conditions.

Not long ago, it was reported that the most commonly used 3D scaffold architectures in cartilage TE were porous 3D sponges [49]. This non-conventional procedure does not allow control over the inner architecture, thus not guaranteeing the desired interconnectivity between pores. Embracing technology advancements, TE overcomes some of the mentioned drawbacks, in particular providing a customized design [50]. Recently, hydrogel scaffolds have been tried [51]. These hydrogels are designed to provide cells with a fully hydrated 3D environment, comparable to the native tissue ECM. However, hydrogels have inadequate mechanical properties that are unfavorable for embedded cells or become too weak for application to the musculoskeletal system. Currently, the 3D hydrogel-enhanced approach provides a foundation to produce biologically and mechanically compatible fabric constructions. Thus, understanding the approaches to make 3D scaffolds (discussed in Section 4) is critical, as they must provide the proper structure to promote homogeneous cell proliferation and/or cell differentiation and, after implantation, host tissue growth. It is important to consider the pore architecture—mechanical and biological properties—strengthening the cellular interconnection. This must be limited within an established design to allow the development of the tissues.

Before the existence of osteochondral TE and the development of centrifugal technology, few artificial materials offered dual bioactivity both for cartilage regeneration and subchondral regeneration [52,53]. As previously mentioned, currently there are several treatments for osteochondral defects (e.g., palliative, restorative, abrasion arthroplasty, chondroplasty, and arthroscopic debridement). Still, they all have limitations, including slower remodeling, the need for post-operation rehabilitation, the possibility of disease transmission, and the possibility of immune reactions. Another strategy that has been used is the transplantation of bioactive materials, such as autologous chondrocytes or cartilaginous tissue, to create defective regions for the osteochondral zone. Several limitations restrict its use, such as the slow maturation of the tissue and long-term recovery [54]. Therefore, it is safe to state that the successful treatment of osteochondral tissue is still a paramount challenge.

In TE, “biomimetics” applies when defining the design and manufacture of scaffolds capable of imitating biological tissue, providing better integration with cartilage and surrounding bone tissue. Osteochondral regeneration, due to the different cartilages and subchondral bone composition, together with biochemical, biomechanical, and biological properties, becomes a big challenge [55]. Therefore, to obtain a biomimetic scaffold it is necessary to provide the different mechanical and biological signals necessary and appropriate to allow the osteochondral regeneration [56]. Over the past decades, there has been a wide variety of technologies for manufacturing 3D scaffolds for osteochondral TE. However, these methods, called traditional methods, are still complex and of low efficiency, since they cannot biologically imitate the native microstructure. To solve these problems, the implementation of AM 3D scaffolds has emerged, which can imitate as closely as possible the anisotropic nature of the ECM and the heterogeneity of the osteochondral tissue [52].

Scaffolds for osteochondral regeneration have several requirements, such as being degradable, biocompatible, as biomimetic as possible, mechanically stable, clinically transferable, and able to be printed and having a viable architecture [57].

### 3.1. Biomaterials

When osteochondral tissue is formed, the subchondral bone acts as a barrier that maintains the integrity of the neocartilage and prevents bone growth in that area, relating to the good regeneration of joint cartilage [58,59]. This interface has been approached in scaffolds through additional layers or phases with different materials and properties. In scaffolds, these phases (between bone and cartilage) have been treated with various porosities [60,61,62], modulus [63,64,65,66], and compositions with materials [67,68] to create the transition between cartilage and subchondral bone [69].

Several designs of scaffolds using synthetic materials have been approached. Given the morphological characteristics in cellular behavior [69,70,71,72,73,74], scaffolds with controlled morphology and chemic should improve osteochondral regeneration, in combination with materials whose properties are known as instructive beings. Accordingly, as demonstrated in Table 1, several types of biocompatible materials have been used to build scaffolds for the osteochondral tissue, including natural polymers, synthetic polymers, metallic polymers, and inorganic polymers [75]. The natural polymers frequently employed are collagen, gelatin, chitosan, alginate, and silk [57]. These have been used in scaffolds for osteochondral regeneration due to their capacity for biomimesis with the ECM structure and good biocompatibility [76,77,78]. The biomaterials that have been exploited for the production of scaffolds are collagen and hyaluronan, among others. Collagen is clearly a natural choice for regenerative medicine or osteochondral TE, since it is the most common protein found in ECM, making up about 90% of the dry weight of articular cartilage [79,80].

The synthetic polymers commonly used for the production of scaffolds for osteochondral regeneration are poly (ɛ-caprolactone) (PCL), poly(lactic acid-coglycolic) (PLGA), poly (ethylene oxide) (PEO), polyglycolic acid, poly (lactic acid) (PLA), PEG, polyactic acid, polydioxanone, and poly(propylene fumarate) [75,91]. These polymers are hydrophobic and can manipulate the properties of the material to achieve the appropriate mechanical behavior. They feature design, 3D composition, and active molecular reactivity groups, at a micro-scale. They have rigidity, elasticity, and porosity, at a macro scale [92]. It is possible to improve hydrophilicity and promote cell fixation by mixing hydrophobic and hydrophilic polymers [75,93,94]. 

Another type of materials that can be utilized are bioceramics, such as HA and tricalcium phosphate (TCP), and bioglass. These are able to stimulate biomineralization for bone regeneration, and the biodegradability of calcium phosphate-based materials can be controlled by changing the calcium/phosphate ratio [75,95,96]. Scaffolds based on bioceramics are bioactive, fragile, and do not resist mechanical stresses. However, they can be packed including natural or synthetic polymers aiming to improve their properties [75,96,97].

Nanoparticles of metals have been extensively studied due to their chemical, physical, and biological properties [96,98]. The unique characteristics of this type of materials are the atoms with high energy that are located in the surface area of the particles [99], the existence of a high relationship between surface area and volume, the high surface energy, and the capacity to store electrons [100]. The metals that have been used are in TE: silver (Ag), gold (Au), iron oxide (IONPs), titanium oxide (TiO_2_), and zinc oxide (ZnO) [101,102,103,104,105,106,107]. In this type of materials, there are several possible applications, such as direct addition of nanoparticles in culture media, use of nanoparticles as a coating, and incorporation of nanoparticles with other materials, such as composites. It all depends on how one intends to develop the materials. Silver nanoparticles show antimicrobial properties, and gold nanoparticles, besides being biocompatible, rarely induce some kind of allergic response. Both can be considered good candidates for biomedical applications. Still, it is necessary to take into consideration their size, shape, stiffness, and surface properties, which are essential for their incorporation into cells [96].

### 3.2. Cell Types for Osteochondral Regeneration

Upon in vivo implantation, MSCs isolated from the bone marrow may form a completely functional bone piece, which provides effective stimuli to attract and allow colonization by hematopoietic precursors [108]. Based on this original premise, MSCs, also from other tissue sources, have been successfully employed in bone regenerative applications. However, experimental findings have documented that very few transplanted MSCs survive following transplantation [109]. 

In the attempt to regenerate the damaged articular cartilage, a crucial challenge is to achieve the morpho-functional regeneration of chondrocytes. The vast majority of approaches to repair or regenerate the articular cartilage are cell-based, aiming to replace functional cells at the injured site. To this aim, either autologous chondrocytes, isolated from unaffected areas, or chondrogenic precursors (mostly MSCs) have been tested. 

Chondrocytes, the only resident cells in the articular cartilage, are metabolically active in charge of ECM production and turnover, by secreting collagens, glycoproteins, proteoglycans, and hyaluronan [110]. Chondrocytes from the different layers of the articular cartilage differ in morphology and size, and, most importantly, in the variety of collagen types, they secrete in the ECM [110,111]. In particular, chondrocytes found in the superficial and mid-zone synthesize ECM components and seem to retain their proliferative capacity, while those of the deep zone are terminally differentiated and exclusively produce ECM. Being entrapped within a dense pericellular matrix and surrounded by an ECM subjected to pressure, chondrocytes retain very limited motility in vivo, even though few studies suggested that chondrocytes can migrate to the site of injury and repair the tissue by synthesizing new ECM [110].

Although current protocols for improved autologous chondrocyte implantation achieve satisfactory clinical outcomes, the extremely low yield of chondrocyte available for the procedure, along with their tendency to dedifferentiate upon in vitro culture amplification, limit the effective utilization of this technique. Besides, long-term studies on chondrocyte implantations reported abnormalities in the histological architecture of the newly formed tissue, showing calcification and irregular fibril bonding in the ECM [112].

As the avascular nature of cartilage tissue hampers the physiological recruitment of stem cells to the site of the lesion, it seemed reasonable to rely on stem cells to aid cartilage regeneration. Various techniques have been developed to heal articular cartilage defects (e.g., drilling, abrasion, microfracture), aimed at enabling new blood supply and forming a path for MSC from the underlying bone niches. MSC expresses a trilineage (osteogenic, chondrogenic, and adipogenic) potential in vitro [113,114]. Culture-expanded MSCs can indeed be induced in vitro toward the chondrogenic lineage and express the morphology and gene signature of chondrocytes [115]. Based on such observations, MSCs have long been considered suitable stem cell sources for cartilage repair and regeneration, also in light of their anti-inflammatory and immunomodulatory effects, potentially beneficial in osteoarthritis-induced degeneration. Nonetheless, it seems that MSCs do not have an effective capability of engrafting upon transplantation and of regenerating a functional hyaline cartilage tissue after articular cartilage has progressed to fibrotic damage [116,117]. MSC progenitors have also been reported to reside within the synovial tissues, but their contribution to cartilage regeneration following disease/damage appears insufficient [117,118]. Other tissue sources for MSC-like progenitors (i.e., adipose tissue and perinatal tissues) have also been successfully differentiated in vitro into chondrocytes. However, MSC transplantation in an experimental protocol of articular cartilage regeneration never proved functionally effective, as, in most cases, fibrocartilage rather than hyaline cartilage is produced in vivo [119].

The rationale for most of the regenerative effects of transplanted MSC appears to reside mostly in the secretion of paracrine factors. Indeed, in early-phase clinical trials, the intra-articular MSC administration induced successful healing and a partial resurfacing of the damaged cartilage, even in the absence of engraftment [120]. The cocktail of bioactive molecules comprised in the MSC secretome includes different growth factors with trophic properties and cytokines that can modulate the inflammatory cascade. This immunomodulatory property makes MSCs able to escape rejection mechanisms long enough to exert their therapeutic action, suggesting their suitability in transplantation [121]. Moreover, they are particularly suitable for treating OA, being able to modulate the inflammatory insult to the synovial membrane, and hence hampering the progression toward fibrosis and osteochondral tissue degeneration [122]. The anti-inflammatory and chondroprotective effects of MSCs from different sources and/or of the secretome derived therefrom, have been proved in several experimental osteoarthritis models and ongoing clinical trials (for a detailed review, see [120]). Of note, MSCs derived from perinatal tissues, such as the human term placenta, exert impressive immunomodulatory properties, including inhibition of T and B cell proliferation and suppression of the inflammatory properties of other immune cells (monocytes, macrophages, dendritic cells, neutrophils, and natural killer cells), along with the induction of immune-regulatory cells (regulatory T lymphocytes and anti-inflammatory M2 macrophages) [123,124]. These properties have laid the foundation for their use for the treatment of inflammatory-based diseases, such as OA. A recent study tested a tissue engineering strategy based on a hybrid scaffold (made of porous polyurethane foam coated under vacuum with calcium phosphates) seeded with human amniotic mesenchymal stromal cells, for osteochondral defect regeneration in a rabbit model. Osteochondral regeneration was partially achieved, especially for the bone component, in the absence of any inflammatory reaction [125].

### 3.3. Additive Manufacturing 

The additive manufacturing (AM) processes have overcome certain problems of conventional methods, such as proper pore size control, design, and interconnectivity. It has enormous potential and exceeds the capabilities of conventional technologies to produce scaffolds with a complex architecture and to achieve an adequate mechanical response to the intended application. The main approaches are fused filament manufacturing (FFD) or fused filament fabrication (FFF), three-dimensional printing (3DP), stereolithography (SLA), and selective laser sintering (SLS). Each process goes through several stages: (i) development of the 3D model through computer drawing (CAD); (ii) the files are stored in standard triangular language (STL), which is a CAD file format that supports 3D printing and computer-aided manufacturing (CAM); and (iii) these files are inserted into the input devices to create 3D models in a layer-by-layer process [92,126]. 

Recently, scaffolds for osteochondral regeneration have used three main types of 3D printing, including SLA, FFF, and SLS [69,127]. Studies have shown that scaffolds with uniform pores for cartilaginous and subchondral regeneration, printed in 3D, with the junction of MSCs demonstrated a chondrogenic and osteogenic differentiation to osteochondral structures [128,129,130]. Table 2 presents the advantages and disadvantages of using these techniques in osteochondral regeneration. It is necessary to decide initially the type of material to be used, since, according to the material, the techniques will be limited.

## 4. Strategies for Osteochondral Regeneration

As already mentioned, it is important to develop a biomimetic scaffold to imitate the gradient of cartilage, calcified cartilage, and bone. Two categories of scaffolds have been developed: biphasic and tri/multiphase [69,75]. Biphasic scaffolds for osteochondral regeneration can be joined by a bone scaffold with a cartilage scaffold [75]. Tri/multi-phase scaffolds demonstrate the ability to biomimic, as closely as possible, the cartilage, bone, and transition zone of the calcified cartilage, constituted in a natural osteochondral bone [135]. Longley et al. [67] demonstrate that while the three-phase approach offers promising results, they need to improve the mechanical properties to withstand biomechanical forces in vivo.

To mimic the characteristics of the osteochondral tissue, it is necessary to produce scaffolds with continuous gradients to induce a smooth transition between the cartilage and the bone component, avoiding instability at the interface [75]. When proposing a scaffold design for osteochondral regeneration, it is important to have a porous structure with a gradient but also with appropriate mechanical properties to match the native tissue. When thinking about the cells, the pore size must be larger, so that they can migrate easily (but not much larger, so that the cells can feel the 3D structure); and the pore size and porosity have significant effects on chondrogenesis and osteogenesis [75]. It is argued that a scaffold to achieve osteogenesis with an improved vascularization requires a porosity of more than 50% and pores slightly larger than 300 µm [136]. To promote chondrogenesis, pores between 90 µm and 120 µm are proposed [137]. This difference in pore size is due to the fact that bone and cartilaginous tissues exhibit different levels of metabolic activity. The oxygen in the cartilage is supplied by the synovial fluid, where chondrocytes consume less oxygen than other cell types. In the subchondral bone, oxygen is supplied by the blood vessels, so the pore size should allow the growth of the blood vessels to exchange nutrients, oxygen, and metabolic waste [138].

O’Reilly et al. [139] developed a computational model to understand the mechanisms by which different TE strategies could improve osteochondral regeneration. They demonstrated that the incorporation of a compact layer in a multiphase scaffold acts as a barrier to angiogenesis, which will provide conditions for the formation of stable cartilage in the defect region. They also point out the importance of angiogenesis control for osteochondral treatment, highlighting that it is important to include MSCs to be successful in osteochondral TE. Finally, they demonstrate the benefit of cell seeding, where only the scaffold was not enough to promote a significantly improved regeneration of the damaged native joint. They concluded that the cell-laden scaffold improves cartilage formation, confining angiogenesis to the bone phase, providing sufficient time for the formation of an overlying layer of stable joint cartilage.

Recently, Nowicki et al. [140] developed a new multiphase scaffold, with different layer geometries and by FFF, to improve the functions of human MSCs from bone marrow (hMSC). They were the first to use a PCL-based form memory material, composed of: PCL-triol, castor oil, and poly(hexamethylene diisocyanate), as osteochondral matrix material. In these layers, nHAs were added, synthesized, and printed. For the cartilaginous layer, chondrogenic growth factors were also manufactured. The results showed better mechanical properties and improved adhesion, growth, and cellular differentiation. These results together with the favorable response of the form memory polymer showed great promise for osteochondral regeneration.

## 5. Conclusions and Future Work

TE offers the possibility of a sustainable and effective treatment against osteochondral defects, where the damaged tissue is replaced by a bi-manufactured long-term replacement tissue. It should be noted that scientists have started to use the biomimetic approach for osteochondral TE, regarding technological, morphological, and structural advances in scaffolds design. Another priority that has been considered is the importance of building a scaffold that mimics ECM at the molecular level.

AM has demonstrated the potential to control several parameters in the design of a scaffold, and consistent production of scaffolds can adapt to each patient. To overcome all the existing gaps that prevent scaffolds for osteochondral regeneration from being optimal, a constant collaborative effort is required among scientists, biologists, biomedical engineers, materials specialists, medical specialists, and clinicians.

In summary, the main challenges are the development of biomimetic and bioactive scaffolds or advanced strategies, which could replicate the native architecture completely and the function of the osteochondral tissue. Thus, a promising scaffold will not only biologically regenerate the osteochondral tissue but also provide a satisfactory postoperative follow-up. Overall, the strategies of structural and biological biomimetism and functionalization will become a deserved focus in osteochondral TE.

## Figures and Tables

**Figure 1 jfb-12-00017-f001:**
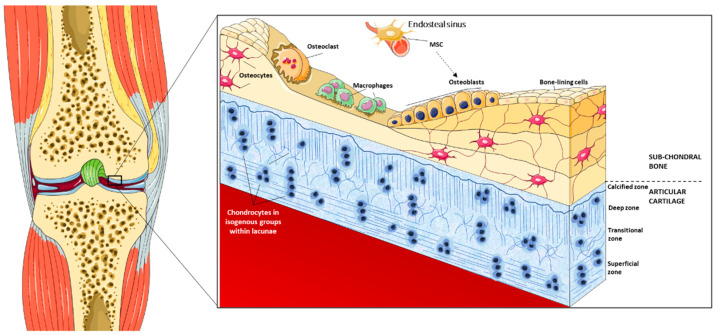
Structural view of the osteochondral boundary. The figure of the left represents a schematic view of the knee joint through a coronal section; the magnification on the right shows an ideal bone-cartilage interface at the femur condyle. Bone tissue is vascularized and houses a wide variety of cells, including perivascular mesenchymal stromal cells (MSCs) that provide mesodermal osteoprogenitors. Osteoblasts are involved in bone matrix deposition and new bone formation; once the process is complete, they rest entrapped within the mineralized matrix and undergo terminal differentiation into osteocytes. Multinucleated osteoclasts, deriving from the blood monocyte lineage, are involved in matrix remodeling, and hence macrophages remove organic matrix debris. The hyaline cartilage features exclusively differentiated chondrocytes, clustered in isogenous groups within ECM lacunae. The cartilage ECM is stratified into layers according to the different orientations of the collagen fibers: in the superficial zone, the fibers run parallel to the articular surface; in the transition zone, the fibers are randomly oriented; in the deep zone, the fibers are perpendicular to the articular surface. The cartilage layer is tightly attached to the bone through its calcified zone. The figure is based on illustrations available at, and modified from, “SMART, Servier Medical Art” (https://smart.servier.com/ (accessed on 4 February 2021)).

**Table 1 jfb-12-00017-t001:** Examples of biomaterials used in osteochondral regeneration.

Biomaterial	In Vitro Studies	In Vivo Studies	Main Results	Ref.
Bacterial cellulose (BC-HA and BC-GAG)	Support attachment and in vitro proliferation of osteoblasts and hACs	Tissue ingrowth and induced no inflammation or immunological reactions after subcutaneous implantation in rats	BC could be modified to mimic two structurally different tissues, and BC exhibited highly desirable biodegradative resorption capability; data presented warrant further extensive and long-term evaluation of BC nanocomposite scaffolds in other animals as well as humans for the eventual translation into clinical practice.	[81]
Collagen matrix with a gradient distribution of low-crystalline HA particles	Ability of cells to be directed by the chondrogenic or osteogenic environment created by the gradient material composition and stiffness in the different regions of the gradient scaffold	Biocompatibility of the gradient scaffold was confirmed by its subcutaneous implantation in rats minimal inflammatory response observed and evidence of cellular differentiation; Scaffold has the potential to selectively differentiate recruited cells	Successful fabrication of a novel composite gradient scaffold with appropriate biomimetic physicochemical and biological properties, for potential application in osteochondral regeneration; Good biocompatibility;	[82]
Poli(N-acriloil 2-glycine) (PACG) and methacrylate gelatin (GelMA) (PACG-GelMA)	High compressive strength (up to 12.4 MPa), and compressive modulus (up to 837 kPa); Incorporating BG could improve the proliferation, ALP activities, and differentiation of hBMSCs, and loading Mn2+ facilitated chondrogenic differentiation of the hBMSCs. Supports cell attachment and spreading, enhances gene expression of chondrogenic-related and osteogenic-related differentiation of human bone marrow stem cells.	Facilitates concurrent regeneration of cartilage and subchondral bone in a rat model.	Superior performance for accelerating cartilage and subchondral bone repair simultaneously in rat knee osteochondral defect	[83]
Two regions: chitosan-hyaluronic acid (cartilage) and chitosan-alginate (bone)	Co-culture with chondrocyte-like (SW-1353 or mesenchymal stem cells) and osteoblast-like cells (MG63), cell proliferation and migration to the interface along with increased gene expression associated with relevant markers of osteogenesis and chondrogenesis		Bilayer scaffold for osteochondral tissue regeneration.	[84]
Alginate-based biphasic scaffold	Good biocompatibility profile	Partial osteochondral regeneration in the rabbit; No evidence of adverse or inflammatory reactions	Limited subchondral bone formation was present, together with a slow scaffold resorption time; Further studies are necessary.	[85]
Horseradish peroxidase (HRP)-cross-linked silk fibroin (SF) cartilage-like layer (HRP-SF layer) + HRP-SF/ZnSr-doped β-tricalcium phosphate (β-TCP) subchondral bone-like layer (HRP-SF/dTCP layer)	Adequate structure, as well as controllable porosity and TCP distribution; Co-culturing of human osteoblasts and human articular chondrocytes showed cell adhesion, proliferation, and ECM production		Complementary in vivo evaluation is necessary to fully validate these structures and confirm the welfare of the ion presence; These hierarchical scaffolds make these constructs encouraging candidates for OC defect regeneration.	[86]
Poly(ε-caprolactone; PCL)	Transwell in vitro culture system of MSC-based constructs enabled the study of soluble biological cues without the influences of mechanical forces, host systemic responses, or animal-to-animal variability that can result in difficulties in interpreting in vivo studies.	Implantation in rat showed that it could significantly shift the phenotype of MSCs from a chondrogenic phenotype to a hypertrophic, osteogenic one.	PCL scaffolds support cellularization as well as extracellular matrix synthesis, accumulation, and remodeling in vivo, regardless of whether or not MSCs were preseeded or precultured;	[87]
Bilayered PLGA/PLGA-HAp	Introduction of the extra cells led to better results	Exhibited satisfactory in vivo efficacy after implanted into the rabbit knee joints for 16 weeks	Potentiality for osteochondral TE, or in situ tissue induction, probably by recruiting the local cells toward chondrogenic and osteogenic differentiation in the porous biomaterials.	[88]
PEGylated poly(glycerol sebacate) (PEGS)	PEGS-12h with low crosslinking degree, hierarchical macro-/micro-porosities, and viscoelasticity significantly enhanced chondrogenic differentiation and cartilage matrix secretion compared to that with high crosslinking degree.	Exhibited extraordinary regenerative efficiency in an articular osteochondral defect model in vivo; Bilayer scaffold reconstructed successfully integrated articular hyaline cartilage and its subchondral bone in 12 weeks		[89]
Silver nanorods (Ag-nr) incorporated wollastonite (CaSiO3)	Cytocompatible, interconnected porous structure and bioactivity of the scaffold.		Healthier substitute for bone tissue engineering compared to other similar materials	[90]

**Table 2 jfb-12-00017-t002:** Advantages and disadvantages of additive manufacturing (AM) processes for osteochondral scaffolds, adapted from references [69,92,131,132,133,134].

Technique	Materials	Advantages	Disadvantages
FFF	Ceramics; Polymers; Thermoplastic.	Speed; Low Cost; Simplicity; Flexibility.	Poor surface quality; Need for heating in the molding process.
SLA	Resins photocurable.	Fast processing times; Good surface finish; Geometrical accuracy; Higher resolution.	Very few materials; Requires support structures; Unable to process functional materials (like metals).
SLS	Polymers; Metals; Alloys; Particles in powder; Ceramics; Stainless steel.	No support structures are required during processing.	Post-processing phase and good surface; Poor mechanical properties.

## Data Availability

Not applicable.

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
