# Peer review of "Challenges and Innovations in Osteochondral Regeneration: Insights from Biology and Inputs from Bioengineering toward the Optimization of Tissue Engineering Strategies"

_jfb, 2021, doi:10.3390/jfb12010017_

Round 1
Reviewer 1 Report
This review article is very well written and summarized the current state-of-the-art TE technology currently being developed for osteochondral regeneration. The authors first provided the deep background of osteochondral tissue, key element/cell signaling needed to be considered for developing osteochondral TE, which is very nice to lead the readers to clearly understand the challenges and ideal properties of osteochondral TE. There are some minor comments and suggestions to enhance the quality of the article.
- Consider changing the title of the article. It is not represented the whole story of the article.
- Check the data/information in Table 2. There are some typos.
- Some statements/sentences need reference support. For example, The natural polymers frequently employed are collagen, gelatin, chitosan, alginate, and silk (page7).
- Consider removing the word "best" in title 4 (page 11) or change to something else to avoid misleading.
- Remove the full name of mesenchymal stem cells (page 12) since it already has been identified as the abbreviation (MSCs).
Author Response
This review article is very well written and summarized the current state-of-the-art TE technology currently being developed for osteochondral regeneration. The authors first provided the deep background of osteochondral tissue, key element/cell signaling needed to be considered for developing osteochondral TE, which is very nice to lead the readers to clearly understand the challenges and ideal properties of osteochondral TE. There are some minor comments and suggestions to enhance the quality of the article.
Authors: Thank you so much for your comments. We do agree that it is necessary to come up with more manuscripts like this, aiming to facilitate further research. We have addressed your comments and suggestions, enhancing its quality.
Consider changing the title of the article. It is not represented the whole story of the article.
Authors: we do agree. We have made a significant change.
Check the data/information in Table 2. There are some typos.
Authors: Dear reviewer, thank you for that alert. We have checked it, as requested.
Some statements/sentences need reference support. For example, The natural polymers frequently employed are collagen, gelatin, chitosan, alginate, and silk (page7).
Authors: Dear reviewer, thank you for that alert. We have included it, as requested.
Consider removing the word "best" in title 4 (page 11) or change to something else to avoid misleading.
Authors: We did have the intent to make the reader think about the best strategies. Accordingly, we have removed the adjective.
Remove the full name of mesenchymal stem cells (page 12) since it already has been identified as the abbreviation (MSCs).
Authors: Dear reviewer, thank you for that alert. We have removed it.
Reviewer 2 Report
The authors presented a review on the state-of-art regarding the challenges, advantages and drawbacks that researchers face regarding tissue engineering for osteochondral regeneration. This manuscript shows an organized work, the results from the available literature being described, but some aspects must be improved before recommending its publication in Journal of Functional Biomaterials.
Please provide more specific the purpose of your review in the abstract and in the body text of the manuscript.
The title includes the word “challenges”, but the authors did not present a special section with challenges in tissue engineering for osteochondral regeneration. I suggest to change the title and to highlight the information described in the manuscript, which is more targeted on the strategies currently used for osteochondral regeneration.
Please check the English language in whole manuscript.
Author Response
The authors presented a review on the state-of-art regarding the challenges, advantages and drawbacks that researchers face regarding tissue engineering for osteochondral regeneration. This manuscript shows an organized work, the results from the available literature being described, but some aspects must be improved before recommending its publication in Journal of Functional Biomaterials.
Authors: Thank you so much for your comments. We do agree that it is necessary to come up with more manuscripts like this, aiming to facilitate further research. We have addressed your comments and suggestions, enhancing its quality.
Please provide more specific the purpose of your review in the abstract and in the body text of the manuscript.
Authors: Dear reviewer, thank you for that suggestion. We have addressed it, accordingly.
The title includes the word “challenges”, but the authors did not present a special section with challenges in tissue engineering for osteochondral regeneration. I suggest to change the title and to highlight the information described in the manuscript, which is more targeted on the strategies currently used for osteochondral regeneration.
Authors: we do agree. We have made a significant change.
Please check the English language in whole manuscript.
Authors: As requested, we have double-checked it for grammar and writing issues.